# `TT-TFHE`: a Torus Fully Homomorphic Encryption-Friendly Neural Network Architecture

**Adrien Benamira**[*]                                                  *adrien.benamira@ntu.edu.sg*
*Nanyang Technological University*
*Singapore*

**Tristan Guérand**[*]                                                  *guer0001@e.ntu.edu.sg*
*Nanyang Technological University*
*Singapore*

**Thomas Peyrin**                                                  *thomas.peyrin@ntu.edu.sg*
*Nanyang Technological University*
*Singapore*

**Sayandeep Saha**                                                  *sayandeepsaha@cse.iitb.ac.in*
*IIT Bombay*
*India*

**Reviewed on OpenReview:** *https://openreview.net/forum?id=tV4ynvae6W*

## Abstract

This paper presents `TT-TFHE`, a deep neural network Fully Homomorphic Encryption (FHE) framework that effectively scales Torus FHE (TFHE) usage to tabular and image datasets using the Truth-Table Neural Networks (`TTnet`) family of Convolutional Neural Networks. The proposed framework provides an easy-to-implement, automated `TTnet`-based design toolbox with an underlying (python-based) open-source Concrete implementation (CPU-based and implementing lookup tables) for inference over encrypted data. Experimental evaluation shows that `TT-TFHE` greatly outperforms in terms of time and accuracy all Homomorphic Encryption (HE) set-ups on three tabular datasets, all other features being equal. On image datasets such as MNIST and CIFAR-10, we show that `TT-TFHE` consistently and largely outperforms other TFHE set-ups and is competitive against other HE variants such as BFV or CKKS (while maintaining the same level of 128-bit encryption security guarantees). In addition, our solutions present a very low memory footprint (down to dozens of MBs for MNIST), which is in sharp contrast with other HE set-ups that typically require tens to hundreds of GBs of memory per user (in addition to their communication overheads). This is the first work presenting a practical solution of private inference (i.e. a few seconds for inference time and a few dozen MBs of memory) on both tabular datasets and MNIST, that can easily scale to multiple threads and users on server side. We further show that in real-world settings, our proposals reduce costs by one to several orders of magnitude compared to existing solutions.

## 1 Introduction

Deep Neural Networks (DNNs) have achieved remarkable results in various fields, including image recognition, natural language processing or medical diagnostics. "Machine Learning as a Service" (MLaaS) is a recent and popular DNN-based business use-case (Philipp et al., 2020; Li et al., 2017; Ribeiro et al., 2015), where clients pay for predictions from a service provider. However, this approach requires trust between the client and

---

[*]Both authors contributed equally to this research

the service provider. In cases where the data is sensitive, such as military, financial, or health information, clients may be hesitant (or are simply not allowed) to share their data with the service provider for privacy reasons. On the service provider's side, training DNNs requires large amounts of data, technical expertise, and computer resources, which can be expensive and time-consuming. As a result, service providers may hesitate to give the model directly to the client, as it may be easily reverse-engineered (or at least make the attacker's task much easier), hindering the growth of MLaaS activity. Allowing the clients to perform the inference locally is also not very practical as any model update would have to be pushed to all clients, not to mention the complex support of the various client hardware/software configurations, etc.

HE/FHE (Gentry, 2009) is an ideal technology to address Privacy-Preserving in Machine Learning (PPML) as it allows the computations to be performed directly on encrypted data. By encrypting its data before sharing it with the service provider, the client ensures that it remains private while the service provider can still guarantee accurate predictions. This solves the trust issue and also gives a competitive advantage in regions where data regulations are stricter, such as Europe's General Data Protection Regulation (GDPR) (Regulation, 2016). The most popular HE schemes are BGV/BFV (Brakerski et al., 2014; Brakerski, 2012; Fan & Vercauteren, 2012), CKKS (Cheon et al., 2017) and Torus-FHE (TFHE) (Chillotti et al., 2016; 2020a) and this paper presents our proposed solution that utilizes TFHE to provide a privacy-preserving MLaaS framework for DNNs. TFHE enables very fast gate bootstrapping as well as circuit bootstrapping and operations over Boolean gates. Moreover, extended versions of TFHE, such as Concrete (Chillotti et al., 2020b), allow programmable bootstrapping and enable evaluation of certain functions during the bootstrapping step itself.

The security and flexibility provided by HE come at a cost, as the computation, communication, and memory overheads are significant, especially for complex functions such as DNNs:

- *Time overhead:* as each mathematical operation required to infer must be executed homomorphically, it is more resource-intensive than non-HE operations. On the server side, an FHE logic gate computation within a TFHE scheme takes milliseconds (Chillotti et al., 2016), compared to nanoseconds for a standard logic gate. On the client side, the only overhead is for the encryption/decryption of the data which is usually not an issue, as milliseconds are enough for those operations.

- *Communication overhead:* the data size required for FHE encryption is significantly larger than that of clear data. An MNIST image sent in clear represents a few kBs, while encrypted within any HE scheme will be of the order of a few MBs (excluding the public key which typically requires hundred(s) of MBs) (Clet et al., 2021). Moreover, PPML schemes not based on TFHE, such as those based on CKKS, require a significant communication overhead due to the need for multiple exchanges between the client and the cloud (Clet et al., 2021).

- *Memory overhead:* TFHE-based schemes typically require a few MBs of RAM on the server side, while those based on CKKS need several GBs per image (Gilad-Bachrach et al., 2016; Brutzkus et al., 2019), even up to hundreds of GBs for the most time-efficient solutions (Lee et al., 2022) (384GB of RAM usage to infer a single CIFAR-10 image using `ResNet`).

- *Technical difficulties and associated portability issues:* mastering an HE/FHE framework requires highly technical and rare expertise. In their survey, Marcolla et al. (2022) delve into the mathematical foundations of HE/FHE schemes, indicating that a strong mathematical background is essential for mastering FHE frameworks. Moreover, knowledge on the respective performances and trade-offs of the different existing implementations is also needed to properly evaluate the feasibility of a HE/FHE based solution.

These challenges can be approached from different directions, but only a few works have considered designing DNNs models that are compatible/efficient with state-of-the-art HE frameworks in a flexible and portable manner. This paper focuses on integrating the recent FHE scheme, Torus-FHE (TFHE), with DNNs.

**Our contributions.** To address the above issues, we propose a DNN design framework called `TT-TFHE` that effectively scales TFHE usage to tabular and large datasets using a new family of Convolutional Neural

Networks (CNNs) called Truth-Table Neural Networks (`TTnet`). Our proposed framework provides an easy-to-implement, automated `TTnet`-based design toolbox that utilizes the Pytorch and Concrete (python-based) open-source libraries for state-of-the-art deployment of DNN models on CPU, leading to fast and accurate inference over encrypted data. The `TTnet` architecture, being lightweight and differentiable, allows for the implementation of CNNs with direct expressions as truth tables, making it easy to use in conjunction with the TFHE open-source library (specifically the Concrete implementation) for automated operations on lookup tables. Therefore, in this paper, we try to tackle the TFHE efficiency overhead as well as the technical/portability issue. We also introduce a novel perspective on PPML by proposing a cost comparison of the various state-of-the-art techniques.

**Our experimental results.** Evaluation on three popular tabular datasets (Cancer, Diabetes, Adult) shows that our proposed `TT-TFHE` framework outperforms in terms of accuracy (by up to +3%) and time (by a factor 7x to 1200x) any state-of-the-art DNN & HE set-up. For all these datasets, our inference time runs in a few seconds, with very small memory and communication requirements, enabling a practical deployment in industrial/real-world scenarios, where tabular datasets are prevalent (Cartella et al., 2021; Buczak & Guven, 2015; Clements et al., 2020; Ulmer et al., 2020; Evans, 2009).

For MNIST and CIFAR-10 image benchmarks, we further explore an approach for private inference proposed by LoLa (Brutzkus et al., 2019), in which the user/client side is able to compute a first layer and send the encrypted results to the cloud. In this real-world scenario, the user/client performs the computation of a standard public layer (such as the first block of the open-source `VGG16` model) and sends the encrypted results to the cloud for further computation using HE. Through experimental evaluation, we demonstrate that our proposed framework greatly outperforms all previous TFHE set-ups (Sanyal et al., 2018; Fu et al., 2021; Chillotti et al., 2021) in terms of inference time. Specifically, we show that `TT-TFHE` can infer one MNIST image in 4.4 seconds with an accuracy of 98.1% or one CIFAR-10 image in 520 seconds with an accuracy of 74%, which is from one to several orders of magnitude faster than previous TFHE schemes, and even comparable to the fastest state-of-the-art HE set-ups (that do not benefit from the other TFHE advantages) while maintaining the same 128-bit security level. To extend our study to larger datasets, we also give projected results on ImageNet (Krizhevsky et al., 2017).

Our solutions represent a significant step towards practical privacy-preserving inference, as they offer fast inference with limited requirements in terms of memory on server side (only a few MBs, in contrary to other non-TFHE-based schemes), and thus can easily be scaled to multiple users. In addition, they benefit from lower communication overhead. In other words, this is the first work presenting a practical solution of private inference (*i.e.* a few seconds for inference time and a few MBs of memory/communication) on both tabular datasets and MNIST. We further strengthen this point by offering a cost comparison with current state-of-the-art approaches. In real-world settings, our proposals reduce costs by one to several orders of magnitude compared to existing solutions.

**Outline.** In Section 2, we present related works in the field of HE and HE-friendly neural networks. A brief introduction to `TTnet` is provided in Section 3. In Section 4, we introduce our proposed `TT-TFHE` framework, while in Section 5 we provide an evaluation of the performance of our framework on various datasets and various privacy settings. Finally, in Section 7, we discuss the limitations of the proposed framework and present our conclusions.

## 2 Related Works

PPML attracted a lot of attention, especially with regard to the implementation of DNNs. Most of these efforts assume that the (unencrypted) model is deployed in the cloud, and the encrypted inputs are sent from the client side for processing. Inference timing of HE-enabled DNN models is the key parameter, but other factors, such as ease of automation and simplicity of such transformation have also been extensively considered (Boemer et al., 2019; Dathathri et al., 2019; Carpov et al., 2015).

Broadly, the problem can be approached from four complementary directions: 1) Optimizing the implementation of some DNN building blocks, such as the activation layers, using HE operations (Jovanovic et al., 2022;

Lee et al., 2022); 2) Parallelizing the computation and batching of images (Gilad-Bachrach et al., 2016; Chou et al., 2018; Brutzkus et al., 2019) (this is aided by the ring encoding of HE in certain cases) and such efforts also include implementing a hybrid client-server protocol for computation (Juvekar et al., 2018; Mishra et al., 2020); 3) Optimizing the underlying HE operations (Chillotti et al., 2016; 2021; Ducas & Micciancio, 2015); 4) Designing a HE-friendly DNN (Sanyal et al., 2018; Lou & Jiang, 2019; Fu et al., 2021). This fourth category has been relatively less explored and is the main focus of this work.

To the best of our knowledge, the FHE-DiNN paper (Bourse et al., 2018) was the first to propose a quantified DNN to facilitate FHE operations. Then, the TAPAS framework (Sanyal et al., 2018) pushed this strategy further by identifying Binary Neural Networks (BNNs) as effective DNN modelling techniques for HE-enabled inference. This direction has been enhanced later by GateNet (Fu et al., 2021), which optimizes the BNN models by grouping the channels to reduce the number of gates. Lately, DCT-CryptoNet (Roy & Roy, 2025) inferred on the frequency of images to reduce memory size and thus reducing inference time.

Yet, none of these works actually explored the automation perspectives for optimizing a model itself for HE inference, as they still heavily leverage some manual optimizations concerning the underlying FHE library. Our proposal `TT-TFHE` is, however, fully automated.

Beyond HE-based approaches, private inference has also been explored through secure Multi-Party Computation (MPC) protocols and GPU-accelerated PPML frameworks. MPC systems such as ABY3 (Mohassel & Rindal, 2018), SecureNN (Wagh et al., 2019), and XONN (Riazi et al., 2019) can achieve low-latency two-party inference and strong correctness guarantees, but require multiple rounds of communication and therefore are less suitable for asynchronous, cloud-only settings. In parallel, GPU-based PPML systems such as CryptGPU (Tan et al., 2021) and DELPHI (Mishra et al., 2020) leverage high-throughput accelerators to significantly reduce latency for arithmetic-heavy privacy-preserving inference, yet depend on specialised hardware and large memory budgets. In contrast, `TT-TFHE` focuses on the non-interactive, single-server setting with an honest-but-curious adversary, aiming for minimal memory footprint, architecture-level automation, and seamless integration with the Concrete FHE ecosystem. These paradigms are therefore complementary: MPC trades communication for speed, GPU-based PPML trades hardware for throughput, while `TT-TFHE` emphasises compactness, non-interactivity, and automated HE-friendly model design.

In addition, compared to previously proposed automated approaches, the translation from non-HE model to HE-enabled model is much simpler as all optimizations are handled during the design phase of the model, making `TT-TFHE` much more amenable for typical machine learning experts with little knowledge of FHE.

## 3 Truth-table DCNN (`TTnet`)

Truth Table Deep Convolutional Neural Networks (`TTnet`) were proposed by Benamira et al. (2024) as DCNNs convertible into truth tables by design, with security applications. While recent developments in DNN architecture have focused on improving performance, the resulting models have become increasingly complex and difficult to verify, interpret and implement. Thus, the authors focused on CNNs, which are widely used, and tried to transform them into Boolean functions that are small enough so that their optimal implementation can be computed practically.

**CNN filter as a Boolean function.** The conversion property of floating CNN weights filter into a binary truth table is achieved by transforming the CNN filter function into a Boolean function. To accomplish this, the complexity of the CNN filter function is reduced by : (A) decreasing inputs size of the CNN filter, (B) using binary inputs and (C) using binary outputs. (A) is achieved by decreasing the number of connections between convolution layers and (B-C) by utilizing the Heaviside step function, denoted as $bin_{act}$, to binarize the features. Please note the model is different from BNN as we preserve real-valued weights.

**CNN filter as an optimized Boolean function.** Their proposed method first converts CNN filters into binary truth tables by 1) decreasing the input size (noted as $n$ in the rest of the paper) which reduces the complexity of the CNN filter function, 2) using the Heaviside step function denoted as $bin_{act} = (1+sgn(x))/2$ (with $sgn$ being the sign function) to transform the inputs and outputs into binary values. This results in a Boolean function stored as a truth table that can be exhausted practically (for $n$ not too large), as

seen in Figure 1a. The optimal implementation of this Boolean function can then be obtained with the Quine–McCluskey algorithm. To achieve high accuracy, the CNN filter must also be non-linear before the step function. Then, the CNN filter becomes a non-linear truth table, which is referred to as a Learning Truth Table (LTT) block.

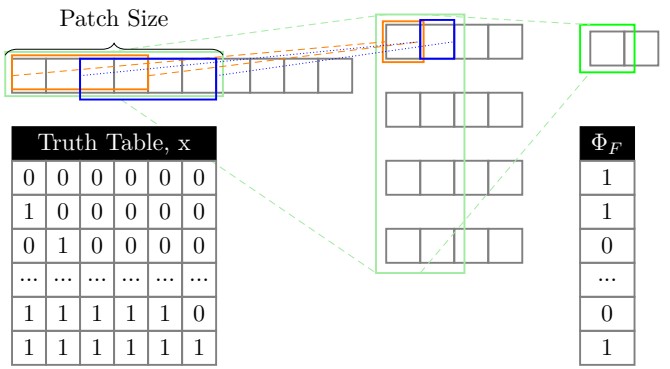

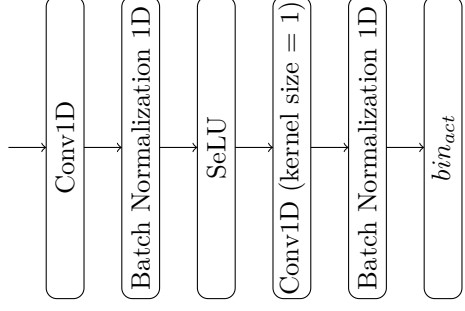

(a) Converting a function $\Phi_F$ into a truth table. The above example has two layers: the first one has parameters (input channel, output channel, kernel size, stride) $= (1, 4, 4, 2)$, while the second $(4, 1, 2, 2)$. The patch size (the patch being the region of the input that produces the feature, which is commonly referred to as the receptive field (Araujo et al., 2019)) of $\Phi_F$ is 6 (i.e., green box) since the output feature (i.e., light green box) requires 6 input entries (i.e., orange and blue box).

(b) LTT overview of a Expanding AutoEncoder LTT in 1 dimension: the Conv1D with kernel size $= 1$ is the amplification layer. The intermediate values are real and the input/output values are binary.

Figure 1: A Learning Truth Table (LTT) block. The intermediate values and weights are floating points, input/output values are binary.

**Description of an LTT block.** Among all the families of LTT blocks possible, we represent in Figure 1b an Expanding Auto-Encoder LTT block (E-AE LTT). An E-AE LTT block is composed of two layers of grouped CNN with an expanding factor. Figure 1a shows the computation of an E-AE 1D LTT block. We can observe that the input size is small ($n = 6$), the input/output values are binary and SeLU is an activation function. Note that while the inputs are binary, the weights and the intermediate values are real. We integrated LTT blocks into `TTnet` as CNN filters are integrated into DCNNs: each LTT layer is composed of multiple LTT blocks and there are multiple LTT layers in total.

**Example: From LTT Weights To Truth Table.** Consider a trained 1D-LTT $\Phi_{\omega_1, \omega_2}$ with input size $n = 4$, a stride of size 1, and no padding. The architecture of $\Phi_{\omega_1, \omega_2}$, given in Figure 1a, is composed of two CNN filter layers: the first one has parameters $\omega_1$ with (input channel, output channel, kernel size, stride) $= (1, 4, 3, 1)$, while the second $\omega_2 = (4, 1, 2, 1)$. The values of the weights ($\omega_1$, $\omega_2$) are given in Figure 1a. The inputs and outputs of $\Phi_{\omega_1, \omega_2}$ are binary, and we denote the inputs as $[x_0, x_1, x_2, x_3]$. To compute the entire distribution of $\Phi_{\omega_1, \omega_2}$, we generate all $2^4 = 16$ possible input/output pairs, and obtain the equivalent truth table. As the inputs are countable, to each input can be associated an output, and we build the table that way. This truth table fully characterizes the behavior of $\Phi_{\omega_1, \omega_2}$. We can then transform the truth table into an optimal Boolean expression using the Quine-McCluskey algorithm (Blake, 1938). This optimal CNF fully characterizes the behaviour of $\Phi_{\omega_1, \omega_2}$ as well and is exactly equivalent.

## 4 The `TT-TFHE` Framework

### 4.1 Threat Model

PPML methods are designed to protect against a variety of adversaries, including malicious insiders and external attackers who may have access to the neural network's inputs, outputs, or internal parameters.

The level of secrecy required depends on the specific application and the potential impact of a successful attack. Common secrecy goals include protecting the input to the inference, ensuring that only authorized parties know the result of the inference, and keeping the weights and biases of the neural network secret from unauthorized parties. Some PPML approaches also aim to keep the architecture of the neural network confidential from unauthorized parties. However, most PPML methods do not address this last point, and some interactive approaches assume that the architecture of the neural network is known to all parties. Moreover, attacks on MLaaS settings exist and are very tricky to defend against (Tramèr et al., 2016; Juuti et al., 2019). Thus, in this paper, we assume that an attacker can access the neural network and only the client's data privacy matters, in an honest-but-curious threat model (Lindell, 2020). It means that the attacker can be located everywhere except at the client's side, and has access to the encrypted inputs, the encrypted outputs, the weights of the model, and does not try to deviate from the designed protocol.

## 4.2 FHE General Set-Up

In this paper, the client $\mathcal{C}$ will encrypt its data locally and send it along with its public key to the server $\mathcal{S}$ (there is no need to send it again once it is pre-shared). The server will compute its algorithm on the encrypted data and send the encrypted result to $\mathcal{C}$, who will decrypt its result locally. The server will have no access to the data in clear.

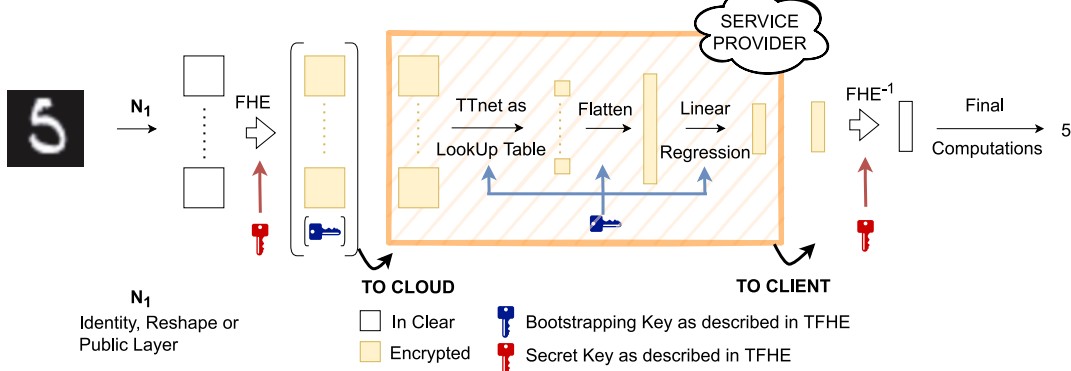

Figure 2: The $N_1/\texttt{TT}$ setting. The client computes locally a layer $N_1$, encrypts the obtained output, and sends it to the server/cloud with the public key (which can be pre-shared). The server will compute through FHE the $\texttt{TTnet}$ layer with the linear regression and send the result to the client. The client can decrypt this output and make a few last computations to obtain the result of the inference. When $N_1$ is identity ($\emptyset/\texttt{TT}$ or more generally $\emptyset/N_2$ for some neural network $N_2$), we denote the setting as Fully Private (Full-Pr).

The classical configuration is that the entire model is computed privately on the server side, and we call this configuration Fully Private (Full-Pr). However, as it is quite hard to defend against model weight-stealing attacks (our threat model does not include model privacy) (Tramèr et al., 2016; Juuti et al., 2019; Carlini et al., 2020), we also consider situations where the client can perform some local pre-processing (*i.e.*, a first layer or block of the neural network) to speed up the server computation without compromising its data security, as introduced by Brutzkus et al. (2019). This setting helps the user to obtain the result of the inference faster than in a Full-Pr situation, with little time/memory cost on their side. Such pre-computation would usually come from a public architecture such as `VGG`, `AlexNet`, or a `ResNet` (Simonyan & Zisserman, 2014; Krizhevsky et al., 2017; He et al., 2016), which are available online. This is a common approach in deep learning, where most models are fine-tuned from one of these three or with fixed first layers followed by a shallow network. In our case, this layer will typically be followed by a `TTnet` model and a linear regression that will be computed through FHE on the server side.

We will denote $N_1/N_2$ a configuration where the client performs the computation of the neural network $N_1$ locally and the remaining part $N_2$ is performed privately on server side. The final linear regression after $N_2$ is always performed privately on server side, but some of the last few computations can be done by the client (for example a part of the sum and the final $ArgMax$ in Podschwadt et al. (2022)). The general setup

$N_1/$TT (where the part performed on server is a TTnet) is depicted in Figure 2. $\emptyset/$TT will represent one extreme case where the entire TTnet neural network is performed privately (Full-Pr) and $N/\emptyset$ will represent the other extreme case where the entire neural network $N$ is performed on client side, except the final linear regression. We will denote $\text{VGG}_{1L}$ the first layer of VGG16 (the first convolution), and $\text{VGG}_{1B}$ its first block (the first two convolutions and the pooling layer afterwards). We finally denote $\epsilon/N$ the setting where the client will perform some normalization, reshape or other small operations on its data locally and then send its encrypted data to the cloud. Table 1 is a recap of the different proposed settings.

Table 1: A recap table of the different proposed settings. A Full-Pr setting refers to a use-case where no normalisation nor reshape is done on the client side. One-hot encoding of categorical variable is acceptable in Full-Pr setting as it does not leak any information about the data or the model.

|  | **In the client** | **In the cloud** |
|---|---|---|
| $\text{VGG}_{1L}/$TT | First layer of VGG | LTT block and Linear Regression |
| $\text{VGG}_{1B}/$TT | First block of VGG | LTT block and Linear Regression |
| $\text{VGG}_{1B}/\emptyset$ | First block of VGG | Linear Regression |
| $\epsilon/$N | Normalization, Reshape | Neural Network |
| Full-Pr or $\emptyset/N$ | | All preprocessing + Neural Network |

### 4.3 Challenges And Optimizations For Integrating TTnet With TFHE-Concrete

The Concrete library (Chillotti et al., 2020b) is a software implementation of TFHE. It is designed to provide a highly efficient and secure platform for performing mathematical operations on Boolean encrypted data. The library utilizes automated operations on lookup tables (concrete-numpy) to achieve high performance while maintaining a high level of security. The library is also designed to be very user-friendly, with simple and intuitive interfaces for performing encryption and decryption operations. The Concrete library has been shown to provide significant improvements in terms of memory and communication overheads compared to other HE schemes. Furthermore, it is open source, which allows researchers and practitioners to easily integrate it into their projects and benefit from its advanced features.

The use of TTnet architecture in combination with TFHE (and more specifically with Concrete) naturally provides a number of advantages. Firstly, the lightweight and differentiable nature of TTnet allows for the implementation of CNNs with direct expressions as truth tables, which is well-suited for Concrete. Additionally, the reduced complexity of the TTnet architecture leads to reduced computations and good scalability. Yet, the integration of TTnet into the TFHE framework presents several challenges that need to be addressed in order to achieve high performance. We detail below the constraints imposed by FHE libraries and the optimizations implemented to overcome these limitations and achieve state-of-the-art performance on various datasets.

**Constraints imposed by Concrete.** The Concrete implementation of the TFHE library, which utilizes automated operations on lookup tables, imposes a maximum limit of 16 on the input bit size $n$ of the truth table. However, $n$ has a strong impact on efficiency (see in Appendix B). Our tests show that $n = 4$ or $n = 6$ seem to offer the best trade-offs. Indeed, since we learn kernels of convolutional layers of size $(3, 2)$ or $(2, 3)$, it is convenient to use $n$ a multiple of 2 and/or 3. Moreover, input sizes larger than $n = 8$ lead to a prohibitive average time per call.

Finally, there is another limitation linked to the precision of the global circuit. Concrete allows for multi-precision circuit, which means that the bitwidth of different parts could be different. For example, if one of the computations operates on 16 bits and another on 4 bits, boths parts will be computed at their respective precisions if needed. This reduces in general the inference times and the memory needs, as expanded in Appendix E. But sometimes this setup cannot be done as cryptographic parameters cannot be found. One can also choose to have the same precision everywhere: but if one of the computations operates on 16 bits, all lookup tables, even small, will require the call time of a 16-bit table lookup. This can be problematic when handling the final linear regression as we will have to sum many Boolean values and the sum result bitwidth might be larger than our planned table lookup size, therefore slowing down our entire implementation. If the

sum result of our linear regression requires 16 bits, all our 4-bit lookup tables call time will be on par with 16-bit table lookups, going from 75ms to almost 5s per call. Therefore, we propose several optimizations to offset a part of the last linear regression to the client, particularly for image datasets (see below).

**Limitations imposed by `TTnet`.** The original `TTnet` paper proposed a training method that is not suitable for high accuracy performance as it was trained to resist PGD attacks (Madry et al., 2017), which reduces accuracy. Additionally, the pre-processing and final sparse layers in `TTnet` being binary, this also leads to a significant decrease in accuracy. To address these limitations, we replace the final sparse binary linear regression with a linear layer with floating point weights (later quantified on 4 bits to not deteriorate too much the performances on Concrete) and propose a new training method that emphasizes accuracy (see below). We also propose to use a setting $N_1/\text{TT}$, where a first layer $N_1$ (a layer or block of a general open-source model) is applied to overcome the loss of information due to the first binarization. This is for example a standard method used in BNNs.

**Training optimizations.** To improve the accuracy of our model, we took several steps to optimize the training process. First, we removed the use of PGD attacks during training, as they have been shown to reduce accuracy. Next, we employed the DoReFa-Net method from Zhou et al. (2016) for CIFAR-10, a technique for training convolutional neural networks with low-bitwidth activations and gradients. Finally, to overcome the limitations of the `TTnet` grouping method, we extended the training to 500 epochs, resulting in a more accurate model.

**Architectural optimizations.** For tabular datasets, the enhancements just proposed are sufficient to achieve high accuracy. However, this is not the case for image datasets such as MNIST, CIFAR-10 and ImageNet. Therefore, we modify the general architecture of our model. Specifically, we use an architecture similar to the one presented for ImageNet in the original `TTnet` paper. The limitation of our model is that it can see far in the spatial dimension but has limited representation in the channel dimension. To balance this property, we use three techniques: multi-headers , residual connections (He et al., 2016), and channel shuffling (Zhang et al., 2017). We have a single layer composed of four different functions in parallel: one LTT block with a kernel size of (3,2) and group 1 (to see far in space and low in channel), one LTT block with a kernel size of (2,3) and group 1, one LTT block with a kernel size of 1 and 6 groups (to see low in space but high in channel representation), and an identity function (as a residual connection for stability). We tried with a second layer to increase accuracy, but this led to sub-optimal performance/accuracy trade-offs with a drastic increase in FHE inference time.

**Client server usage as optimizations.** We describe here a solution to the multi-precision issue: we utilize the client's computing resources not only to prepare the input to the server, but also to post-process the server's output. Namely, the client will compute a small part of the final linear regression. Indeed, we quantify the weights of the final linear regression of `TTnet` to 4 bits, which we divide into 4 binary matrices. Then, the server performs partial sums on each of these 4 matrices. Since the outputs of `TTnet` are binary, the weights are binary, and the maximum number input bitwidth for our lookup tables is 4 bits. For optimal performance, we perform sub-sums of size 16 to ensure that the result of each sub-sum is always lesser than $2^4$ as proposed by Zama's team[1]. It is important to note that by doing this, we maintain the privacy of the weights of the linear regression as they remain unknown to the client. Additionally, the function computation by the client is very light, for example in the case of the MNIST dataset in the $\text{VGG}_{1B}/\text{TT}$ setting, it will represent at most $\frac{N_{\text{features}}}{N_{bits}} = \frac{576}{16} = 12$ sums of 4-bit integers to be performed for each 4 weight matrices. The client will eventually add these four outputs to obtain its final result, *i.e.* computing $\sum_{i=0}^{3} 2^i * w_i * \text{output}_i$. Also, note that this client computation is fixed and will remain the same even if the model needs to be updated. Finally, this limitation introduces a trade-off between communication costs and operative costs: we chose here to work with the "cheaper" solution, accordingly to the philosophy of our work to reduce memory costs, therefore by splitting the final linear regression only when the cryptographic parameters are not found. Appendix E goes into more details on the costs difference between the two setups.

---

[1]https://community.zama.ai/t/load-model-complex-circuit/369/4

# 5 Results

The project implementation was done in Python, with the PyTorch library (Paszke et al., 2019) for training, in Numpy for testing in clear, and the Concrete library v2.10.0[2] for FHE inference. Our workstation consists of 4 Nvidia GeForce 3090 GPUs (only for training) with 24576 MiB memory and eight cores Intel(R) Core(TM) i7-8650U CPU clocked at 1.90 GHz, 16 GB RAM. For all experiments, the CPU Turbo Boost is deactivated and the processes were limited to using four cores.

## 5.1 Tabular Datasets

Table 2: Tabular dataset results for `TT-TFHE` and competitors. All our models use a table lookup bitwidth of $n = 5$, except for Diabetes where we use $n = 6$. We emphasize that the various experiments have been conducted on a different number of CPU cores. Normalized results to a single CPU core are denoted with an ⋆.

| | **FHE family** | **#CPU cores** | Adult | | | | Cancer | | | | Diabetes | | | |
|---|---|---|---|---|---|---|---|---|---|---|---|---|---|---|
| | | | Full-Pr | | $\epsilon$/TT | | Full-Pr | | $\epsilon$/TT | | Full-Pr | | $\epsilon$/TT | |
| | | | Acc. | Time | Acc. | Time | Acc. | Time | Acc. | Time | Acc. | Time | Acc. | Time |
| Jovanovic et al. (2022) | CKKS | 1⋆ 64 | 81.6% | 7.5h 420s | - | - | - | - | - | - | - | - | - | - |
| TAPAS (Sanyal et al., 2018) | | 1⋆ 16 | - | - | - | - | 97.1% | 56s 3.5s | - | - | 54.9% | 66m 250s | - | - |
| XGBoost 6bits (Zama) | TFHE | 1⋆ 4 | 86.0% | 64s 16s | - | - | 96.4% | 14.4s 3.6s | - | - | 57.9% | 111.6s 27.9s | - | - |
| **Ours** | | 1⋆ 4 | 85.3% | 356.4s 89.1s | 85.3% | 22.4s 5.6s | 97.1% | 7.6s 1.9s | 97.1% | 4.32s 1.08s | 57.0% | 72.64s 18.16s | 57.0% | 4.8s 1.2s |

Our experimental results shown in Table 2 demonstrate the superior performance of the proposed `TT-TFHE` framework on three tabular datasets (Cancer, Diabetes, Adult) in terms of both accuracy and computational efficiency. The framework achieved an increase in accuracy of up to +3% compared to state-of-the-art DNN & HE set-ups based on TFHE such as TAPAS (Sanyal et al., 2018) or on CKKS such as the recent work from Jovanovic et al. (2022). More impressively, the inference time per CPU was significantly reduced by a factor 1200x, 13x, and 825x on Adult, Cancer, and Diabetes datasets respectively on the $\epsilon$/TT setting and by a factor 75x, 7x, and 55x on Adult, Cancer, and Diabetes datasets respectively on the Full-Pr setting. Compared to XGBoost, XGboost gives better accuracy on a similar bandwidth than our models. Our framework is faster in both Full-Pr and $\epsilon$/TT setting except for the Adult dataset.

This enables the practical deployment of our framework in industrial and real-world scenarios where tabular datasets are prevalent (Cartella et al., 2021; Buczak & Guven, 2015; Clements et al., 2020; Ulmer et al., 2020; Evans, 2009), with low memory and communication overhead (see Table 4 in Section 5.3). Note that these results are either in the Full-Pr or $\epsilon$/TT setting, as the binarization process in `TT-TFHE` would depends on learned parameters for continuous variables. In Full-Pr setting, the continuous variables are quantized, the client data is then encrypted and sent to the server. In the $\epsilon$/TT setting, the continuous variables are binarized with the parameter learned, reshaped to fit `TTnet` input and then encrypted and sent to the cloud.

## 5.2 Image Datasets

Our experimental results and comparisons for image datasets are given in Table 3.

### 5.2.1 Fully Private (Full-Pr) and $\epsilon/N$ Settings

In the Full-Pr configuration, we focused on the performance of our method on the MNIST dataset, as the binarization process in `TT-TFHE` resulted in a significant loss of accuracy for CIFAR-10 or an increase in inference time. `TT-TFHE` offers a competitive trade-off compared to other TFHE-based methods, such as TAPAS (Sanyal et al., 2018), GateNet (Fu et al., 2021), Zama (Chillotti et al., 2021; Stoian et al., 2023) or DCT-CryptoNets (Roy & Roy, 2025). It is three orders of magnitude faster than TAPAS or GateNet, while

---

[2]https://docs.zama.ai/concrete/2.10

Table 3: Image dataset results for `TT-TFHE` and competitors. Results denoted with ⋆ are estimated by the original authors, not measured. All our models use a table lookup bitwidth of $n = 4$, except the underlined ones that use $n = 6$. Experiments in this Table have been conducted on a different number of CPU cores. Table 12 in Appendix gives the normalized results to a single CPU core.

| | | Full-Pr (∅/N) | | | | | $\epsilon/N$ | | VGG$_{1B}$/∅ | VGG$_{1L}$/N | | VGG$_{1B}$/N |
|---|---|---|---|---|---|---|---|---|---|---|---|---|
| **TFHE-based schemes** | | TAPAS | GateNet | Zama | Zama | Zama | DCT-CryptoNets | **Ours** | **Ours** | Zama | **Ours** | **Ours** |
| #CPU cores | | 16 | 2 | 6 | 8 | 8 | 96 | 4 | 4 | 128 | 4 | 4 |
| **MNIST** | Acc. (%) | 98.6 | 98.8⋆ | 97.1 | 97.6 | 98.7 | - | 97.2 | 97.5 | - | 98.2 | 98.1 |
| | Time | 37h | 44h⋆ | 115s | 35min | 84min | - | 54.02s | 0.04s | - | 8.7s | 4.4s |
| **CIFAR-10** | Acc. (%) | - | 80.5⋆ | - | - | 87.5⋆ | 91.6 | - | 70.4 | 62.3 | 69.4/72.1 | 74.1/75.3 |
| | Time | - | 3920h⋆ | - | - | 5h⋆ | 22.3m | - | 0.4s | 29m | 8.7m/1h | 8.7m/1h |

| | | $\epsilon/N$ | | | | | Full-Pr (∅/N) |
|---|---|---|---|---|---|---|---|
| **non-TFHE-based schemes** | | CryptoNets | Fast CryptoNets | Lola | Lee *et. al.* | Rovida *et al.* | SHE |
| #CPU cores | | 4 | 6 | 8 | 1 | 1 | 10 |
| MNIST | Acc. (%) | 99 | 98.7 | 99.0 | - | - | 99.5 |
| | Time | 4.2m | 39s | 2.2s | - | - | 9.3s |
| CIFAR-10 | Acc. (%) | - | 76.7 | 74.1 | 91.3 | 91.53 | 92.5 |
| | Time | - | 11h | 12.2m | 37.8m | 4.3m | 37.6m |

showing only an accuracy reduction of 1.4%. In comparison to Zama, our method is 3x faster per CPU for the same level of accuracy. Additionally, we highlight that for our single layer LTT block of size $n = 6$, we require 1600 calls to 6-bit lookup tables, which leads to an inference time of only 54 seconds on four CPU cores. Only DCT-CryptoNet manages to reach a high accuracy on CIFAR-10: they first convert the images into the frequency domain which allows smaller inputs and apply various small operations before encrypting the inputs and sending them to the cloud. Our solution is still faster, at the cost of accuracy.

`TT-TFHE` is even competitive in terms of inference time with some non-TFHE-based schemes such as (Fast) CryptoNets (Gilad-Bachrach et al., 2016), but can be one order of magnitude slower with slightly lower accuracy. Therefore, one can observe that the Full-Pr setting of TFHE implemented in our framework generally underperforms compared to the very latest Full-Pr CKKS or leveled FHE scheme such as SHE. This is explained by the first binarization process in `TT-TFHE`, which compresses too much information embedded in the input image. Yet, we again emphasize the many advantages of TFHE-based solutions compared to non-TFHE-based ones: little memory required allowing easy/efficient multi-client inference, low communication overhead, no security warning on TFHE while CKKS secret key can be recovered in polynomial time (Li & Micciancio, 2021) (a fix was proposed afterwards by Li et al. (2022) but not yet implemented in SEAL for example), etc. Furthermore, CKKS focus on amortized runtime through SIMD which makes comparison difficult. Moreover, We will see in the next sub-section that the performance situation is very different in the setting where the client can perform some pre-computation layer.

### 5.2.2 Other Settings

We first observe that when allowing the client to apply a simple pre-processing layer, the performance increases drastically for `TT-TFHE`. We have implemented and benchmarked both VGG$_{1L}$/TT and VGG$_{1B}$/TT settings, both with `TTnet` models with $n = 6$ and $n = 4$ and we obtained a $7\times$ performance improvement, with an increase in accuracy. For reference, we have also tested the VGG$_{1B}$/∅ setting where only a linear regression is computed privately on server: we remark that adding a `TTnet` in the server computation indeed improves accuracy by about 4%.

One could argue that more `VGG` blocks could be computed on client side to further increase the accuracy, but this would reduce the generality of the first layers and lead to PPML solutions that would not adapt very well to multiple use cases. We have tried blocks of other more recent CNNs than `VGG`, such as `DenseNet` (Huang et al., 2017), but the results remained very similar.

We can compare the `TT-TFHE` results to some TFHE-based competitors, as Zama proposed a similar setting[3] with $\mathtt{VGG}_{1L}$ pre-computed by the client for CIFAR-10, and against which we infer 100x faster per CPU and with a 10% increase in accuracy.

Our `TT-TFHE` results are now even competitive against non-TFHE-based solutions (even though they again miss many of TFHE advantages), being faster than (Fast) CryptoNets (Gilad-Bachrach et al., 2016) and SHE (Lou & Jiang, 2019), and on par with Lola (Brutzkus et al., 2019) and Lee et al. (2022) per CPU. We note that SHE and Lee *et. al.* have better accuracy than our model.

### 5.2.3 Preliminary Results On ImageNet

We would like to highlight that ImageNet is not included in our result tables as it remains a very challenging task to perform in an industrial setting. Our estimation suggests that achieving a top-1 accuracy of 21% would take around 120 days with `TTnet`, which is obviously not practical in real-life scenarios. Furthermore, other methods such as LoLa (Brutzkus et al., 2019) or Lee et al. (2022) did not mention ImageNet in their work. In SHE (Lou & Jiang, 2019), the authors projected a time of 165 days for a fully homomorphic encryption (FHE) model and around 5 hours on 10 cores for a leveled model. However, it is important to note that their experiments were conducted on a 1TB machine, which is not readily available in most industrial settings. Only Roy & Roy (2025) managed to reach 66.3% accuracy in almost 2h, on 96 threads requiring 4.21GB. While our paper aims to provide an industrial approach, ImageNet results are almost out of the scope of our work due to the high computation time and resources required. Designing `TTnet` architectures compatible with high-resolution images (e.g., $224 \times 224$) would require many more layers of table lookups and large kernel sizes, thereby compounding cost. In future work, we propose to investigate hierarchical table-splitting strategies, GPU/FPGA acceleration of TFHE lookup bootstraps, and mixed HE/MPC pipelines to enable ImageNet-scale encrypted inference under `TT-TFHE`.

### 5.3 Memory/Communication Cost Of `TT-TFHE`

In Tables 4 and 5, we give the memory and communication needs for all our settings. We can observe that the deeper the representation, the smaller the communication needs: the inputs to the server become smaller as we go deeper into the neural network (there are also fewer computations to do in the server). Some settings do not need public keys as only a linear regression is performed, and thus no programmable bootstrapping is involved (Chillotti et al., 2021). Then, the largest the lookup tables (in terms of the number of features and size), the larger will be the public keys as there will be more bootstrapping. Also, the optimization proposed in Section 4.3 to ease the linear regression comes with a cost: it increases the size of the outputs. Indeed, only the $\mathtt{VGG}_{1B}/\emptyset$ setting does not use this optimization as there is no lookup-table involved and thus no bootstrapping. Moreover, the encrypted inputs size increases with the number of features. Finally, the pre-processing on the client side is also to take into account: between the Full-Pr and the $\epsilon/\mathtt{TT}$ settings on the Adult dataset, the memory needed is increased by 1000x because of the binarization done with $\leq$ operations on the data. This increase can also be observed on the Diabetes dataset with a 313x increase for the same reason. Therefore, an $\epsilon/\mathtt{TT}$ setting would be preferable for real-life use as memory requirements and inference time are much lower.

### 5.4 Comparison With Other Methods.

As stated in Clet et al. (2021), CKKS solutions usually require a much larger communication and memory cost than TFHE ones. In Table 6, we compare between each method the amount of RAM needed on server side for CIFAR-10 dataset. CryptoNets, SHE, Lola and Lee *et al.* are either Full-Pr or $\epsilon/N$, whereas Zama and ours are not as we let the user do one `VGG` layer or one `VGG` block locally. We measured the RAM used by Zama on the same machine used for our experiments. We observe that we use less memory than every proposed method with a competitive accuracy. Lee et al. (2022), Rovida & Leporati (2024), SHE (Lou & Jiang, 2019) and Roy & Roy (2025) outperform our accuracy by more than 17%. But with huge memory cost: indeed we only need 47.5 MB which is 8000× less than Lee *et al.*. SHE did not report their RAM needs, but they used a 1 TB machine to run their experiments.

---

[3]`https://github.com/zama-ai/concrete-ml/tree/release/0.6.x/use_case_examples/cifar_10_with_model_splitting`

Table 4: Memory and communication needs for `TFHE` in various settings. Each dataset column is split into two configurations: Full-Pr and $\epsilon$/`TT`. Communication cost includes encrypted inputs, outputs, and public keys. All tabular models use a table lookup bitwidth of $n = 5$ and image models a bitwidth of $n = 4$, except underlined values which use $n = 6$.

| | | Adult | | Cancer | | Diabetes | |
|---|---|---|---|---|---|---|---|
| | | Full-Pr | $\epsilon$/`TT` | Full-Pr | $\epsilon$/`TT` | Full-Pr | $\epsilon$/`TT` |
| **Client** | Encryption Keys | 135 kB | 38.1 kB | 22 kB | 22 kB | 135 kB | 21.6 kB |
| | Public Keys | 1.56 GB | 220.0 MB | 168.96 MB | 168.96 MB | 1.53 GB | 101.6 MB |
| | Encrypted Input Size | 12.5 MB | 8.5 MB | 1.2 MB | 1.2 MB | 37 MB | 4.6 MB |
| | Encrypted Output Size | 0.25 MB | 0.03 MB | 0.5 MB | 0.5 MB | 0.4 MB | 2.0 MB |
| **Server** | RAM | 15 GB | 13.6 MB | 1.04 MB | 1.04 MB | 6.9 GB | 3.4 MB |
| **Communication Cost (with key)** | | 1.6 GB | 229.0 MB | 170.66 MB | 170.66 MB | 1.6 GB | 108.2 MB |
| **Communication Cost (without key)** | | 13 MB | 8.53 MB | 1.7 MB | 1.7 MB | 38 MB | 6.6 MB |

Table 5: Memory and communication needs for `TT-TFHE` in various settings. The client will have to send the public keys and the encrypted inputs to the server and then receive the encrypted outputs from it. The communication cost is therefore the sum of these three items. All tabular models use a table lookup bitwidth of $n = 5$ and images model a table lookup bitwidth of $n = 4$, except for underlined results who use a bitwidth of $n = 6$.

| | | MNIST | | | | CIFAR-10 | | |
|---|---|---|---|---|---|---|---|---|
| | | $\epsilon/N$ | $\text{VGG}_{1B}/\emptyset$ | $\text{VGG}_{1L}/\text{TT}$ | $\text{VGG}_{1B}/\text{TT}$ | $\text{VGG}_{1B}/\emptyset$ | $\text{VGG}_{1L}/\text{TT}$ | $\text{VGG}_{1B}/\text{TT}$ |
| **Client** | Encryption Keys | 70.9 kB | 10 kB | 21.7 kB | 21.7 kB | 10 kB | 21.7 kB / 334.41 kB | 21.7 kB / 334.41 kB |
| | Public Keys | 766.9 MB | 0 MB | 152.4 MB | 152.4 MB | 0 MB | 538.41 MB / 14.79 GB | 538.41 MB / 14.79 GB |
| | Encrypted Input Size | 100 MB | 11.5 MB | 18.4 MB | 9 MB | 75.7 MB | 484.24 MB / 3.31 GB | 484.24 MB / 3.31 GB |
| | Encrypted Output Size | 2.5 MB | 0.1 MB | 40.6 MB | 640.3 MB | 0.1 MB | 160.08 MB / 2.5 MB | 160.08 MB / 2.5 MB |
| **Server** | RAM | 53.7 MB | 0.6 MB | 18.1 MB | 18.1 MB | 0.5 MB | 18.4 MB / 47.5 MB | 18.4 MB / 47.5 MB |
| **Communication Cost (with key)** | | 870 MB | 11.6 MB | 131 MB | 161 MB | 76.3 MB | 1.3 GB / 21.5 GB | 1.3 GB / 21.5 GB |
| **Communication Cost (without key)** | | 102.5 MB | 11.6 MB | 29 MB | 10 MB | 76.3 MB | 1.2 GB / 18.3 GB | 1.2 GB / 18.3 GB |

Table 6: RAM usage between different methods for the inference of a single image of CIFAR-10. CryptoNets authors did not report results on CIFAR-10, but LoLa team (Brutzkus et al., 2019) estimated that it would take around 100 GBs to infer one image of this dataset with CryptoNets. SHE memory is reported as an upper bound as numbers were not given except for the experimental setup. The DCT-CryptoNets authors also did not publish memory requirements, but we obtained them through direct correspondence.

| Dataset | Method | FHE type | Accuracy | Server RAM |
|---|---|---|---|---|
| CIFAR-10 | CryptoNets | BFV | - | 100 GB |
| | SHE | LTFHE | 92.5% | <1 TB |
| | LoLa | BFV | 74.1% | 12 GB |
| | Lee *et al.* | CKKS | 91.31% | 384 GB |
| | Rovida *et al.* | CKKS | 91.53% | 15 GB |
| | DCT-CryptoNet | TFHE | 91.6% | 4.21 GB |
| | Zama $\text{VGG}_{1L}/N$ | TFHE | 62.31% | 8.3 GB |
| | **Ours** $\text{VGG}_{1L}/\text{TT}$ | TFHE | 69.4% | 18.4 MB |
| | **Ours** $\text{VGG}_{1B}/\text{TT}$ | TFHE | 74.1% | 47.5 MB |

The best accuracy is the `ResNet` proposed by DCT-CryptoNet Roy & Roy (2025), but it also leads to a high consumption in RAM. With LoLa setting, the accuracy is indeed lower but it requires 32x lesser RAM than Lee et al. (2022). Then, our method reduces again the memory on server by almost a factor of 252x for the same accuracy. Rovida & Leporati (2024) manage to reach a similar accuracy than DCT-CryptoNet Roy & Roy (2025), but with a higher memory requirement. RAM size on server matters for cloud computing as pricing increases along with memory needs[4], thus low-memory solutions help the scalability of the MLaaS.

---

[4]https://aws.amazon.com/lambda/pricing/

# 6 Cost Study And Comparisons For Deployment At Industrial Scale

We propose a novel perspective on the costs of FHE for industrial-scale machine learning deployment. While accuracy is important, time and memory are equally crucial from an industrial standpoint as they have an important impact on costs.

From the viewpoint of cloud providers, the costs are determined by the computation and RAM memory per image (assuming a full load of the servers). We aim to rank the methods based on the costs required to deploy FHE-based private machine-learning technologies in real-life scenarios.

To achieve this, we make some assumptions: we compare the cost per method for 100,000 customers, with each customer sending one input request. We used the AWS Lambda pricing of \$0.0000166667 GB-second [5]. We do not consider the cost of communication (this is to our disadvantage as CKKS-based solutions typically require a much larger communication cost than TFHE ones).

We conduct a comparison of two realistic use cases for large-scale deployment: tabular and MNIST datasets. For each case, we calculate the cost of a sample image.

## 6.1 Tabular Dataset

Table 7 gives a comparison of the costs of two different methods for the tabular datasets. As memory requirements were not disclosed in previous works by Jovanovic et al. (2022) or Sanyal et al. (2018), we could not include them in the comparison. Yet, to compare our method with state-of-the-art methods, we implemented the Zama library's open-source Pythonic XGBoost. We tested 2 levels of model precision for the XGboost: 4-bit and 6-bit precision. The cost gain factor in Table 7 is a measure of the relative improvement in cost efficiency of the proposed method compared to our TTnet model (calculated as the ratio of the two costs per sample). For instance, in the case of the 4bit XGBoost model on the Adult dataset, the cost gain factor of our proposed method over Zama's XGBoost is 250. This means that our proposed method achieves a 250-fold reduction in the cost per sample compared to Zama's XGBoost.

Table 7: Comparison of cost per sample for different methods across datasets. The time and memory needed are given per sample. The cost is estimated for 100,000 customers. All the Zama XGBoost models are $\epsilon/N$ as a quantization step in the clear is necessary for them, ours are also $\epsilon/N$ because a binarization step with a learned parameter is done.

| Methods | Adult | | | Cancer | | | Diabetes | | |
|---|---|---|---|---|---|---|---|---|---|
| | XGboost | | Ours | XGboost | | Ours | XGboost | | Ours |
| Precision model | 6bits | 4bits | 5bits | 6bits | 4bits | 6bits | 6bits | 4bits | 6bits |
| Accuracy | 86.0% | 84.3% | 85.3% | 96.4% | 94.3% | 97.1% | 57.9% | 57.9% | 57.0% |
| FHE Time (4 cores) | 16s | 12.8s | 5.6s | 3.6s | 2.3s | 1.08s | 27.9s | 26.9s | 1.2s |
| Memory | 372MB | 335MB | 3.4MB | 329MB | 294MB | 12MB | 381MB | 349MB | 32MB |
| Total Cost | \$8.9 | \$7.9 | \$0.03 | \$2.0 | \$1.1 | \$0.02 | \$17.7 | \$15.6 | \$0.06 |
| Cost gain factor | 281 | 250 | 1 | 62 | 35 | 1 | 558 | 493 | 1 |

Table 7 shows that our proposed method achieves significantly lower costs than the state-of-the-art methods. In particular, on the Adult dataset, our method achieves a total cost of only \$0.03, which is between $250\times$ to $281\times$ lower than that of Zama's XGBoost. Moreover, our method achieves comparable accuracy while requiring significantly less memory and FHE time. These results demonstrate that our proposed method is a more cost-efficient and practical solution for large-scale deployments for tabular datasets. We also explored implementing another compact classical ML model, GLRM (Wei et al., 2019), using a naive quantization of its final linear parameters. However, this resulted in a substantial loss of accuracy. Adapting GLRM, or any other rule-based model, to the quantized setting would require extensive modifications, which fall outside the scope of this work.

---

[5]https://aws.amazon.com/lambda/pricing/

## 6.2 MNIST Dataset

Table 8 presents a comparison of different methods on the MNIST dataset. The table displays performance metrics, including accuracy, FHE inference time (4 cores), and memory usage, as well as the estimated cost per sample and the cost gain factor. The memory requirements were not disclosed in the works of Sanyal et al. (2018). We also excluded SHE from the comparison, as its memory requirement of 1TB RAM makes it impractical for scalability.

Our proposed method achieved an accuracy of 98.1%, with an FHE time of 4.4s and a memory usage of 18MB. Our method has a total cost of only $0.13, which is between $114\times$ to $36000\times$ lower than other state-of-the-art methods. Our method is largely more cost-effective than the other methods for the MNIST dataset.

Table 8: Comparison of cost per sample for different methods on the MNIST dataset. The time and memory needed are given by sample. The cost is estimated for 100,000 customers.

| Methods | Performances | | | Costs | |
|---|---|---|---|---|---|
| | Accuracy | FHE Time (4 cores) | Memory | Cost Total | Cost Gain Factor |
| Zama | 97.1% | 172.5s | 1.5GB | 431$ | 3.3k |
| Fast CryptoNets | 98.7% | 59s | 48GB | 4720$ | 36k |
| Lola | 99.0% | 4.4s | 2GB | 14.7$ | 114 |
| Ours | 98.1 % | 4.4s | 18MB | 0.13$ | 1 |

# 7 Limitations And Conclusion

## 7.1 Limitations

Our proposed framework is still not as performant as the latest non-TFHE-based solutions with regard to running time and accuracy. Moreover, CIFAR-10 and larger datasets remain out of reach for industrial use. Furthermore, while `TT-TFHE` enhances privacy during inference (the server never receives decrypted client data and computes only on ciphertexts), it is important to acknowledge that inference-time privacy does not eliminate all risks. For example, model inversion, side-channel leakage (e.g., timing or memory access patterns), or collusion between client and server remain potential vulnerabilities. Furthermore, the model architecture may be exposed in our setup, so adversarial reconstruction of inputs from outputs remains an open concern. As such, deployment of `TT-TFHE` in sensitive settings should consider complementary mitigations (such as differential-privacy preprocessing, audited hardware environments, or side-channel hardening) beyond simply encrypting the inference pipeline.

## 7.2 Conclusion

In this paper, we presented a new framework, `TT-TFHE`, which greatly outperforms all TFHE-based PPML solutions in terms of inference time, while maintaining acceptable accuracy. Thanks to the compact nature of `TTnet`, our proposed framework is a practical solution for real-world applications, particularly for tabular data and small image datasets like MNIST, as it requires minimal memory/communication cost, provides strong security for the client's data, and is easy to deploy. We believe that this research will spark further investigations into the utilization of truth tables for privacy-preserving data usage, a technology advocated by the recent NIST Artificial Intelligence Risk Management Framework (AI, 2023).

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

## A    General architecture of `TTnet`

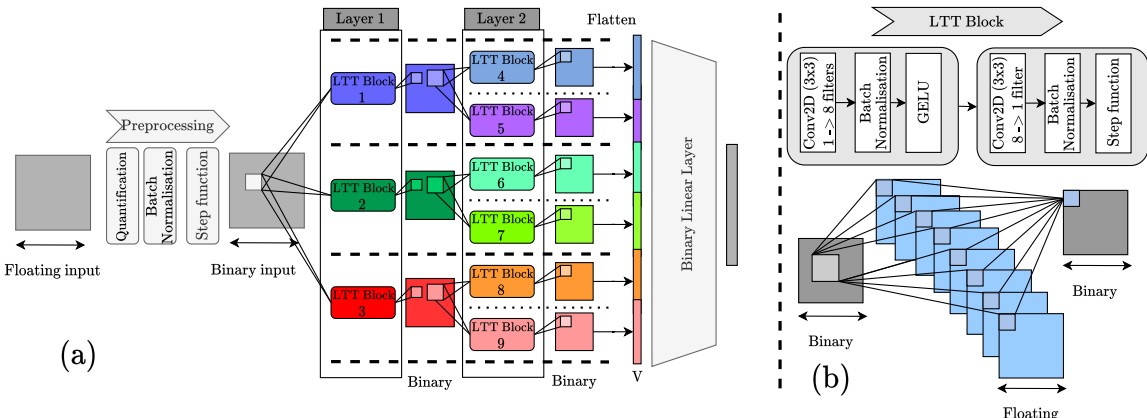

Figure 3: **(a)** General architecture of the `TTnet` model with a one-channel input. Layer 0 is a pre-processing layer that allows image binarization. Then follow two layers of Learning Truth Table (LTT) blocks: three blocks in the first layer, six in the second. It should be noted that the LTT block of layer 2 does not take as input all the filters of layer 1, as it is usually the case: it only takes the filter of their groups. Finally, the last linear layer performs the classification. **(b)** Architecture of a LTT block. It is composed of two layers of grouped 2D-CNN with an expanding factor of 8. It can be seen as an expanding auto-encoder. The intermediate values are real and the input/output values are binary.

## B    On Concrete table lookups

Table 9 presents the average time per call on lookup tables of different sizes for Concrete. We can observe that computation time doubles between 5-bit and 6-bit tables. From 8-bit to 9-bit, it increases by a factor 3.8x with almost 3 seconds for each call. Thus, we focused on tables with a maximum size of 6 bits. The code used to obtain this table is available on Zama website[6]. Experiments were performed on our CPU, without Turbo Boost.

## C    Architecture description

We detail below the architecture of the various models we use. All the linear regression weights are quantified to 4-bit.

**Adult.**   This model is composed of one LTT block of kernel size 5 and stride 4 with no padding. It results in 274 rules being activated, *i.e.* with weight during linear regression being different from 0.

**Cancer.**   This model is composed of one LTT block of kernel size 5 and stride 4 with no padding. It results in 80 rules being activated, *i.e.* with weight during linear regression being different from 0.

**Diabetes.**   This model is composed of one LTT block of kernel size 6 and stride 5 with no padding. It results in 295 rules being activated, *i.e.* with weight during linear regression being different from 0.

**MNIST - $\epsilon$/`TT`**   This model is composed of one LTT block of kernel size 6 and stride 2 with no padding. The input is binarized and resized to $20 * 20$ before entering the LTT block. It is followed by a linear layer of 1600 features to 10 classes.

---

[6]https://docs.zama.ai/concrete-numpy/getting-started/performance

Table 9: Measured time of a lookup table call through TFHE library Concrete, according to the table input bit sizes.

| Input bit size | Average time per call (ms) |
|:---:|:---:|
| 1 | 49.3 |
| 2 | 57.6 |
| 3 | 57.3 |
| 4 | 74.6 |
| 5 | 75.2 |
| 6 | 169.9 |
| 7 | 353.4 |
| 8 | 774.4 |
| 9 | 2979.5 |
| 10 | 2756 |
| 11 | 3023.2 |
| 12 | 3732.5 |
| 13 | 3956.5 |
| 14 | 4030.1 |
| 15 | 4009.4 |
| 16 | 4499.5 |

**MNIST - $\text{VGG}_{1B}/\emptyset$**   This model is composed of the first $\text{VGG}$ block followed by a linear layer of 1176 features to 10 classes.

**MNIST - $\text{VGG}_{1L}/\text{TT}$**   This model is composed of the first $\text{VGG}$ block followed by one LTT block of kernel size 2 and stride 1 with no padding and 24 channels. It is followed by a linear layer of 1176 features to 10 classes.

**MNIST - $\text{VGG}_{1B}/\text{TT}$**   This model is composed of the first $\text{VGG}$ block followed by one LTT block of kernel size 2 and stride 1 with no padding and 16 channels. It is followed by a linear layer of 576 features to 10 classes.

**CIFAR-10 - $\text{VGG}_{1B}/\emptyset$**   This model is composed of the first $\text{VGG}$ block followed by a linear layer of $64*11*11 = 7744$ features to 10 classes.

**CIFAR-10 - $\text{VGG}_{1L}/\text{TT}$**   There are two models for this setup:

- The first model is composed of the first $\text{VGG}$ layer followed by three LTT blocks in parallel of kernel size $(2,2)$ and stride 1 with no padding and one residual layer in parallel. The outputs are then concatenated into a vector of size $64*11*11*4 = 30976$. It is followed by a linear layer of 30976 features to 10 classes.

- The second model is composed of the first $\text{VGG}$ layer followed by three LTT blocks and one residual layer all in parallel. The first LTT block uses a kernel size $(3,2)$ and stride 2 with no padding, the second one a kernel size $(2,3)$ and stride 2 with no padding, the third one a kernel size of size 1 and 6 groups and stride 2 with no padding. The outputs are then concatenated into a vector of size $64*11*11*4 = 30976$. It is followed by a linear layer of 30976 features to 10 classes.

**CIFAR-10 - $\text{VGG}_{1B}/\text{TT}$**   There are two models for this setup:

- The first model is composed of the first $\text{VGG}$ block followed by three LTT blocks in parallel of kernel size $(2,2)$ and stride 1 with no padding and one residual layer in parallel. The outputs are then concatenated into a vector of size $64*11*11*4 = 30976$. It is followed by a linear layer of 30976 features to 10 classes.

- The second model is composed of the first `VGG` block followed by three LTT blocks and one residual layer all in parallel. The first LTT block uses a kernel size $(3, 2)$ and stride 2 with no padding, the second one a kernel size $(2, 3)$ and stride 2 with no padding, the third one a kernel size of size 1 and 6 groups and stride 2 with no padding. The outputs are then concatenated into a vector of size $64 * 11 * 11 * 4 = 30976$. It is followed by a linear layer of 30976 features to 10 classes.

## D  Dataset description

All datasets have been split 5 times in a 80-20 train-test split for k-fold testing.

**Adult.**  The Adult dataset comprises 48,842 individuals, each with 14 features and a label that indicates whether their income is above or below 50K\$ USD or not. After one hot encoding, it resulted in 94 binary features and 6 numerical features. The dataset is available at `https://archive.ics.uci.edu/ml/datasets/Adult`.

**Cancer.**  The Cancer dataset is composed of 569 data points, each with 30 numerical characteristics. The objective is to predict whether a tumor is malignant or benign. To achieve this, we transformed each integer value into a one-hot vector, resulting in a total of 81 binary features. The dataset is available at `https://archive.ics.uci.edu/ml/datasets/Breast+Cancer+Wisconsin+(Diagnostic)`.

**Diabetes.**  The Diabetes dataset includes 100,000 patient records, each with 50 characteristics that are both categorical and numerical. We retained 43 of these features, 5 of which are numerical, and the rest are categorical. This resulted in 291 binary features and 5 numerical features. The goal is to predict one of the three labels for hospital readmission. The dataset is available at `https://archive.ics.uci.edu/ml/datasets/Diabetes+130-US+hospitals+for+years+1999-2008`.

## E  Ablation Study on CIFAR10

In order to evaluate the effectiveness of our proposed framework, we conducted an ablation study on the CIFAR10 dataset. Our study investigated the impact of architectural optimizations on the number and precision of truth tables, the precision of the last layer, and the error precision of the FHE model. The resulting table of evaluation metrics (Table 10) displays the accuracy, time on 4 cores, and memory usage for various configurations of the framework. Our baseline model was trained with 4-bit truth table precision, 4-bit LT precision, 48 filters, and error = 0. Preliminary results suggest that accuracy can decrease with less than 4 bits, and time increases dramatically with higher precision. These findings provide important insights into the optimization of our framework for practical applications in real-world scenarios.

Table 10:  Ablation study on the CIFAR10 dataset. The table displays the evaluation metrics for various configuration of the proposed `TT-TFHE` framework. The baseline model was trained with 4-bit truth table precision, 4bit LT precision and error =0. The estimated results are denoted with an asterix (*).

|  | Parameters | Accuracy | Time on 4 cores | Memory |
|---|---|---|---|---|
| Influence Truth Table precision | 4bit | 74.1% | 522s | 18 MB |
|  | 6bit | 75.3% | 1h | 48 MB |
|  | 16bit | 80.2% | 5 days* | * |
| Influence LR precision | 1bit | 65.8% | 253s | 80 MB |
|  | 4bit | 74.1% | 522s | 18 MB |
|  | 8bit (splitted) | 74.1% | 483s | 1.6 GB |
| Influence p error (on first 100 images) | 0 | 82 | 522s | 18MB |
|  | 0.05 | 82 | 437s | 14MB |
|  | 0.1 | 82 | 353s | 14MB |

In Table 11, we give the different metrics with a part of the LR being delegated to the client or not (split vs. not split). We can observe that while the inference time reduces greatly ($1.6\times$), the RAM needed is a few order of magnitude higher due to the large outputs being computed at that time.

Table 11: Impact of splitting the linear regression on 4bits on all the different metrics on the CIFAR-10 dataset, in the $\texttt{VGG}_{1B}/\texttt{TT}$ setting.

|  |  | CIFAR10 | |
|---|---|---|---|
|  |  | Not Split | Split |
| **Client** | Encryption Keys | 33 kB | 22 kB |
|  | Public Keys | 538 MB | 107 MB |
|  | Encrypted Input Size | 485 MB | 485 MB |
|  | Encrypted Output Size | 160 kB | 1.6 GB |
| **Server** | RAM | 18 MB | 1.6 GB |
|  | Inference Time | 520s | 320s |
| **Communication Cost (with key)** |  | 1 GB | 2.3 GB |
| **Communication Cost (without key)** |  | 485 MB | 1.6 GB |

## F  Single CPU estimations

Table 12: Normalized image dataset results for $\texttt{TT-TFHE}$ and competitors to a single CPU core. Results denoted with $\star$ are estimated by the original authors, not measured. All our models use a table lookup bitwidth of $n = 4$, except the underlined ones that use $n = 6$.

| TFHE-based schemes | Full-Pr ($\emptyset/N$) | | | $\epsilon/N$ | | $\texttt{VGG}_{1B}/\emptyset$ | $\texttt{VGG}_{1L}/N$ | | $\texttt{VGG}_{1B}/N$ |
|---|---|---|---|---|---|---|---|---|---|
|  | TAPAS | GateNet | Zama | DCT-CryptoNets | **Ours** | **Ours** | Zama | **Ours** | **Ours** |
| #CPU cores | 1 | 1 | 1 | 1 | 1 | 1 | 1 | 1 | 1 |
| **MNIST** Acc. (%) | 98.6 | 98.8* | 97.1 | - | 97.2 | 97.5 | - | 98.2 | 98.1 |
| Time | 592h | 88h* | 11.5m | - | 4m | 0.16s | - | 34.8s | 17.6s |
| **CIFAR-10** Acc. (%) | - | 80.5* | - | 91.6 | - | 70.4 | 62.3 | 69.4/72.1 | 74.1/75.3 |
| Time | - | 7840h* | - | 36h | - | 1.6s | 61.8h | 35m/4h | 35m/4h |

| non-TFHE-based schemes | $\epsilon/N$ | | | | | Full-Pr ($\emptyset/N$) |
|---|---|---|---|---|---|---|
|  | CryptoNets | Fast CryptoNets | Lola | Lee *et. al.* | Rovida *et. al.* | SHE |
| #CPU cores | 1 | 1 | 1 | 1 | 1 | 1 |
| **MNIST** Acc. (%) | 99 | 98.7 | 99.0 | - | - | 99.5 |
| Time | 16.6m | 3.9m | 17.6s | - | - | 93s |
| **CIFAR-10** Acc. (%) | - | 76.7 | 74.1 | 91.3 | 91.53 | 92.5 |
| Time | - | 66h | 97m | 37.9m | 4.3m | 6.3h |

