# OpenReview forum: "TT-TFHE: a Torus Fully Homomorphic Encryption-Friendly Neural Network Architecture"
_TMLR — Accepted by TMLR_

### Review · Reviewer_tLL8 · 2025-08-05

**Summary Of Contributions:**

This paper presents TT-TFHE, a framework for running deep neural network inference on encrypted data using the TFHE scheme. The key idea is to design convolutional layers as truth tables (via a family of networks called TTnet), which aligns well with TFHE’s lookup-table-based computation. The authors integrate this approach into a PyTorch + Concrete implementation and demonstrate results on both tabular and image datasets.

**Audience:**

Yes

**Broader Impact Concerns:**

No concerns.

**Claims And Evidence:**

Yes

**Requested Changes:**

1.	It would be helpful to know more about the training process, how long it takes, whether there are specific difficulties in convergence, and how stable the training is with the changes authors made (e.g., DoReFa, longer epochs)?
2.	The authors acknowledge in the limitations that performance on harder vision tasks isn’t competitive with the very best HE schemes. That’s fine, but it might help to also reflect on this tradeoff earlier in the discussion and suggest future work or directions to improve accuracy (possibly adding more layers or smarter quantization?).
3.	Typos:
Convolutionnal -> Convolutional
infered -> inferred

Awkward/redundant phrasing: We also give projected results on ImageNet, We try to tackle the TFHE efficiency overhead as well as the technical/portability issue, Therefore, in this paper, we try to tackle…
A pass over the writing to remove redundancy and tighten some phrasing would improve flow.
4.	It would be interesting to briefly discuss whether TT-FHE can be considered in multi-party/federated settings, as this could broaden the practical applications of the framework.

**Strengths And Weaknesses:**

Strengths:

1.	The paper focuses on making FHE-based inference practical, which is a timely and important problem.
2.	The integration of TTnet with TFHE is interesting, and the framework is implemented in a way that’s accessible and efficient.
3.	The authors evaluate on a mix of tabular and image datasets and include comparisons with other HE methods.
4.	The cloud deployment cost analysis is particularly valuable and appreciated.
5.	Compared to CKKS or BFV-based approaches, this method uses significantly less memory, making it more suitable for real-world multi-user applications.

Weaknesses:
1.	Performance on CIFAR-10 is decent but doesn’t quite reach the best results in the literature (especially those using leveled HE or hybrid schemes).
2.	The paper rightly admits that ImageNet-scale use cases are out of scope for now, but that does limit generalizability.
3.	The paper mentions using DoReFa-Net, removing PGD, and extending to 500 epochs, but doesn’t give much detail on how training proceeds, how long it takes, or how stable it is.
4.	Some parts, particularly in the abstract and introduction, are a bit wordy or repetitive.

---

> ### Author Response · Authors · 2025-09-02
> **Answer to reviewer tLL8**
>
> Thank you for your review of our work, we appreciate your feedback and will integrate your comments into our final version.
>
> We would like to address the following points:
>
> 1. It would be helpful to know more about the training process, how long it takes, whether there are specific difficulties in convergence, and how stable the training is with the changes authors made (e.g., DoReFa, longer epochs)?
> —> it takes around a day to train on Cifar10, and a week on ImageNet. The training is stable thanks to the Straight Through Estimator. The training converges at around epoch 150 and then the last epochs are grapping the last few accuracy %.
>
> 2. The authors acknowledge in the limitations that performance on harder vision tasks isn’t competitive with the very best HE schemes. That’s fine, but it might help to also reflect on this tradeoff earlier in the discussion and suggest future work or directions to improve accuracy (possibly adding more layers or smarter quantization?).
> —> We agree to surface the trade-off earlier and propose concrete avenues to improve accuracy within TFHE/Concrete constraints, such as PTQ for the LTT layers, more layers (but will increase inference time and memory), different truth table size for different layers (mixed precision), etc. as a future direction. As scaling TTnet to ImageNet would be a whole another project, we keep this as another project.
>
> 3. Typos: Convolutionnal -> Convolutional infered -> inferred Awkward/redundant phrasing: We also give projected results on ImageNet, We try to tackle the TFHE efficiency overhead as well as the technical/portability issue, Therefore, in this paper, we try to tackle… A pass over the writing to remove redundancy and tighten some phrasing would improve flow.
> —> we will remove redundant wording
>
> 4. It would be interesting to briefly discuss whether TT-FHE can be considered in multi-party/federated settings, as this could broaden the practical applications of the framework.
> —> we will start first by saying that we are not multi party experts. As far as we know, multi party private inference infers on a circuit, which makes TT-TFHE a good candidate to try it on. Regarding federated learning, as TTnet is differentiable it should also work.
>
> Please let us know if it addresses your remarks.

---

### Review · Reviewer_WL3m · 2025-08-10

**Summary Of Contributions:**

The papers presents an approach for training a variant of CNN known as TTNet on encrypted data using Torus Fully Homomorphic Encryption. This is done using the Torus FHE Concrete Library, which provides the necessary capabilities for implementing FHE computations. The key contribution claimed is an "efficient implementation of the TTNet Model using the TFHE-Concrete library"

**Audience:**

No

**Claims And Evidence:**

No

**Requested Changes:**

Please refer to the strengths and weaknesses section.

**Strengths And Weaknesses:**

The main contribution of the work is an implementation (reportedly effective) of a special type of Convolutional Neural Network (which is conducive to FHE) using a user-friendly library (TFHE-Concrete).
The authors discuss a few different challenges that seemingly arise in such an implementation:

A. Constraints imposed by the library: FHE is inherently sensitive to computational depth - this is due to noise accumulation in FHE computations. This results in different trade-offs between performance and efficiency. It appears that Concrete is no different and the authors tuned a certain trade-off parameter (similar to hyper parameter tuning) in terms of input bit size. While an implementation challenge, it is not a noteworthy research challenge, imo.

B. Limitations imposed by TTNet: The original paper that proposed TTNet had the objective of resisting PGD attacks. The authors propose changes in architecture to remove this constraint. In the words of the authors such a change -- "is for example a
standard method used in BNNs"

C. Training Optimizations: In the words of the authors: "... to overcome the limitations of the TTnet grouping method, we extended the training to 500 epochs, resulting in a more accurate model." Training longer for better accuracy is the most obvious "Training Optimization".

D. Architectural optimizations: The authors explore a bunch of other proposed architectures (multiple headers, residual connections, shuffling etc.) which are commonplace in deep networks literature. I fail to see any novelty here as well.

As written, I find that the paper lacks sufficient novelty.

The paper is reasonably well written and easy to follow. However there are some rather strange statements in the paper. A few examples:

1. "Technical difficulties and associated portability issues: mastering an HE/FHE framework requires
highly technical and rare expertise." I am not sure if a research paper can be accepted just because it is hard to implement and that the authors posses a unique skill. It doesn't strengthen the paper in anyway and I suggest such statements be removed (there is no way to verify it in any case - that implementing FHE based approaches is a rare skill).
2. "… so that their optimal implementation can be computed practically" What does computing an implementation mean exactly?
3. "… propose a new training method that emphasizes accuracy" ..  isn’t that pretty much all the training methods objective? the goal is to improve accuracy.
4. "…Finally, to overcome the limitations of the TTnet grouping method, we extended the training to 500 epochs, resulting in a more accurate model" --> this cannot be claimed as novelty, I am sure.

---

> ### Author Response · Authors · 2025-09-02
> **Answer to Reviewer WL3m**
>
> Thank you for your review of our work, we appreciate your feedback and will integrate your comments into our final version.
>
> We would like to address the following points:
>
> Our work should be read as a systems contribution for privacy-preserving inference: we make TT-style CNNs run efficiently under Concrete/TFHE by:
> 1. calibrating bit-width and precision jointly with the bootstrapping
> 2. transforming the architecture where necessary to reduce PBS precision while preserving the Booleanizable operator set
> 3. empirically evaluating TTnet modules under TFHE cost, reporting accuracy/latency/memory trade-offs. We will revise the paper to de-emphasize any generic DL novelty claims and focus on these FHE-specific, measurable contributions.
>
> Our goal aligns with TMLR’s criteria: we provide clear evidence for our claims and target an audience in FHE/PPML systems. As TMLR notes, novelty per se is not required for acceptance; we therefore emphasize the soundness and interest of our FHE-specific pipeline and findings.
>
> To our persepctive, the novelty does not come from the execution of combining TFHE with TTnet (which we believe is sound and complete), but from the observation that TTnet-like models are a perfect fit for Concrete/TFHE. In addition, our results are novel as they represent the state-of-the-art for PPML using TFHE in terms of performance. Finally, TTnet + Concrete/TFHE is the first published method that is a fully practical and production-ready solution for private inference, with low memory and low communication contraints.
>
>
> Regarding the comments:
>
> 1. “Technical difficulties and associated portability …”
> —> we will remove this comment
>
> 2. “ … so that their optimal implementation can be computed practically”
> —> we are referring here to the size of the truth tables. From Quine McCluskey algorithm, it is NP hard to get the optimal boolean function from a truth table. Therefore, we need to keep the table size low in order to be able to get the equivalent boolean function in a reasonable time. This goes well with the increased time in computation for bigger bit size in concrete.
>
> 3. “propose a new training method that emphasizes accuracy" .. isn’t that pretty much all the training methods objective? the goal is to improve accuracy.
> —> the PGD training aims to increase verifiable accuracy, which also lowers accuracy in many cases as it is aimed for robustness, see [1].
>
> We are not claiming this as a novelty, as it is classic but as a methodological change compared to the original TTnet work.
>
> 4. “…Finally, to overcome the limitations of…”
> —> we are not presenting this as a novelty, but merely a method change from the original work. We will remove this statement as it could be confusing.
>
>
> Once again, thank you for your honest feedback on our work.
> Please let us know if it answers your remarks.
>
>
>
> [1] Robustness May Be at Odds with Accuracy, ICLR 2019 https://arxiv.org/pdf/1805.12152

---

> > ### Comment · Reviewer_WL3m · 2025-10-24
> > **thank you**
> >
> > I thank the authors for their responses to the feedback.

---

### Review · Reviewer_sape · 2025-09-05

**Summary Of Contributions:**

TT-TFHE focuses on tabular and image datasets using TFHE and truth-table neural networks (TTnet). TT-TFHE is a Pytorch and Concrete based framework providing an automated way of scaling TTnets using TFHE. TT-TFHE outperforms other FHE implementations based on BFV or CKKS. In their evaluations, the authors report increased accuracy and significant runtime improvements (7-1200x) for three popular datasets.

For MNIST and CIFAR-10, the authors focus on the approach by LoLa which splits the inference and has the client compute the first layer and send the encrypted results to the cloud. It’s not clear though, if the runtime improvements they report are using this method and compare against frameworks that do end-to-end FHE evaluation.

The related works section is very short and fails to report and compare with multiple notable FHE works. Many works from venues like CCS, USENIX, S&P, NDSS, PETS, IEEE TIFS, etc, which focus on PPML inference (even LLM inference!!), look-up tables (LUTs) both in the context of FHE and MPC, FHE inference using GPUs, and many more have not been included here. For example, a notable work from 2021 is CrypTen, that focuses on MPC-based PPML inference and also provides a very easy to use framework using PyTorch. Not to mention since then, how many works have improved upon CrypTen. I understand that the authors want to focus on CNNs and LTTs, but the SOTA on PPML has improved a lot compared to what the authors report.

Unfortunately, this is also reflected in Section 5, where the comparisons are only against 2-3 works. More elaborate (and fair) comparisons need to be made both theoretically and experimentally.

I’d appreciate a bit more details on TTnets, I am not sure I really understood how CNN filters can be converted and optimized to Boolean functions. Is there any novelty on that or is this just from previous works? If it’s the latter, citations are missing.

On the threat model, the authors mention malicious insiders, but this is incompatible with FHE’s threat model, which assumes an HBC CSP. The threat model needs to be cleaned up and explained further. The authors say “in this paper, we assume that an attacker can access the neural network and only the client’s data privacy matters.” This is incorrect. A malicious CSP can completely corrupt the computation. The threat model is very simple and the same as most FHE works: it’s a semi-honest CSP. This section is confusing. Also, it fails to address who has the inputs and who owns the model.

In section 4.2 it became clear that the client computes the first layer. Why? This reveals weights to the client, if the model owner is different. There have been works that do this approach as well, but I also don’t think they are cited here. Some examples that come to my mind: Zama, PermLLM, Fission, Split learning for health. But all these are kind of insecure, see: Model inversion attacks against collaborative inference, Testing Robustness of Homomorphically Encrypted Split Model LLMs, and Unleashing the tiger: Inference attacks on split learning. Since the authors take a “split inference” approach, they’d need to demonstrate why this approach is secure and not vulnerable to such attacks.

Is your framework open source?

**Audience:**

Yes

**Claims And Evidence:**

No

**Requested Changes:**

Please address my comments W1, W2, and W3, in that order of priority.

W1 and W2 are crucial to be able to assess the paper better.

W3 is crucial to understand whether TT-FHE is secure against known attacks.

**Strengths And Weaknesses:**

S1: The TTnet approach seems very interesting and novel.
S2: I like the fact that the authors created an easy to use framework that can be used and help new research.

W1: The related works section is very short and skips multiple related works in the PPML area. There have been GPU-based FHE works that will probably be significantly faster (and that's fine!), MPC works that evaluate LLMs, and even other FHE works, that have not been mentioned here. It's fine if TT-TFHE is slower than related works, but they need to be referenced, otherwise the comparisons are unfair.
W2: I'd also like to see comparisons with related works experimentally. (To clarify, W1 is about theoretical, W2 is about experiments).
W3: This approach seems to share ideas from split inference/split learning works, which have been vulnerable to attacks. Is this framework also vulnerable to these attacks?
W4: Is your framework open source?

---

> ### Author Response · Authors · 2025-09-16
> **Answer to Reviewer sape**
>
> We would like to first thank the reviewer for his comments and his appreciation of our work.
>
> We would like to adress the following points:
>
> - "For MNIST and CIFAR-10, the authors focus on the approach by LoLa .... "
> —> We detailed the way of inferencing in Table 1. All the subsequent results tables are using these notations. Some of our results are end to end, and some uses LoLa’s setup.
>
> - "The related works section is very short and ... "
> —> We understand your point, but we really focused on FHE based PPML solutions as PPML is a very large field with very different methods. Otherwise, we would have also explored using TTnet for MPC but it would have been a whole other paper, and much longer and maybe less clear. We can add a paragraph on MPC methods and explain more our position in the PPML field.
>
> - "I’d appreciate a bit more details on TTnets, ... "
> —> We built this work upon [1], that we referenced in part 3: Truth table DCNN . The main idea is to binarize the inputs of the CNN filter, which makes the input countable. Therefore, we can convert the whole filter/kernels into lookup tables. More optimisations are made in [1] in order to make it scalable (work on the activation function and the gradients for example).
>
> - "On the threat model, the authors mention malicious insiders, but ... "
> —> We disagree with you on this point. If the computation is corrupt, this is not an issue as the privacy of the client is still enforced. The inputs are owned by the client, and the model by another entity. The setup is explained in Figure 2.
>
> - "In section 4.2 it became clear that the client computes the first layer. ..."
> —> Zama is cited as [2]. PermLLM and Fission do not use FHE, and Split learning for health use Federated Learning which makes them out of scope for our work. We chose to use public weights with a first VGG16 layer, as it is a popular first layer for many vision neural networks. We propose different results for different kinds of setup.
>
> - "But all these are kind of insecure, see: ..."
> —>  this is a good concern, but would be out of scope of our work. We will cite these works to warn on possible model inversion attacks, but we think that there is a trade off there.
> W2: I'd also like to see comparisons with related works experimentally. (To clarify, W1 is about theoretical, W2 is about experiments).
> —> we ran all concrete based FHE work locally. We asked authors when the details where either unavailable or not feasible for us to run on our machines.
>
> W3: This approach seems to share ideas from split inference/split learning works, which have been vulnerable to attacks. Is this framework also vulnerable to these attacks?
> —> we do not do split learning
>
> W4: Is your framework open source?
> —> we can give the code on demand basis
>
>
> Thank you again for your remarks and feedbacks on our work.
>
>
> [1] Adrien Benamira, Thomas Peyrin, Trevor Yap, Tristan Guérand, and Bryan Hooi. Truth table net: Scalable,
> compact & verifiable neural networks with a dual convolutional small boolean circuit networks form. In
> Kate Larson (ed.), Proceedings of the Thirty-Third International Joint Conference on Artificial Intelligence,
> IJCAI-24, pp. 13–21. International Joint Conferences on Artificial Intelligence Organization, 8 2024. doi:
> 10.24963/ijcai.2024/2. URL https://doi.org/10.24963/ijcai.2024/2. Main Track
>
> [2] Andrei Stoian, Jordan Frery, Roman Bredehoft, Luis Montero, Celia Kherfallah, and Benoit Chevallier-Mames.
> Deep neural networks for encrypted inference with tfhe. arXiv preprint arXiv:2302.10906, 2023.

---

> > ### Comment · Reviewer_sape · 2025-10-07
> > **Response**
> >
> > Dear authors,
> >
> > In my review, I asked to address my comments W1, W2, and W3, in that order of priority. W1 and W2 are crucial to be able to assess the paper better. You only briefly touched upon the related works comment (W1 and W2). You responded that they only focused on FHE-based PPML, but then, at least, I would expect to see comparisons against GPU-and-FHE-based PPML works, which you didn't comment on. Additionally, I think comparisons against MPC-based PPML works are crucial to better understand the efficiency of this work. MPC-based PPML tends to be significantly faster than FHE, so choosing not to compare is not fair.
> >
> > Finally, you seem to have misunderstood my comment about the threat model. Achieving correctness vs privacy against active adversaries are two different things. Claiming this work is secure against malicious adversaries is wrong. I am also not fully satisfied with the responses to the rest of my comments, but they are not as important.

---

### Decision · Action_Editor_JSJS · 2025-10-25

**Recommendation:** Accept with minor revision

**Additional Comments:**

The reviewers raised some valid points, some of which have already been addressed in the revised version. Despite the reviewers' perspectives being mixed, I believe this paper offers a valuable contribution to FHE-ML, a promising area that has yet to realise its full potential.

Authors must address the following issues as part of the camera-ready preparation for final approval and publication. My comments should be considered *in conjunction with* those provided by the reviewers, and both sets of comments (mine and the reviewers') should be addressed in the camera-ready version, if not already done.

a) Threat model: Clarify whether TT-TFHE assumes an honest-but-curious server and remove any implication of malicious-adversary security as suggested by one of the reviewers (unless you would like to argue otherwise in terms of the validity of this statement). Also, please explain this aspect in more detail, both in the paper and in your response to these issues when submitting the camera-ready version.

b) Comparative context: Add a concise paragraph situating TT-TFHE relative to MPC and GPU-based PPML methods, explaining the scope difference even without new experiments, although some experiments would be strongly recommended.

c) Avoid overstatements such as “fully production-ready” or “first practical FHE inference,” and correct typographical errors (e.g. Convolutionnal -> Convolutional in the abstract). There are more typos; please read the manuscript carefully while revising.

d) Briefly explain how truth-table CNN filters are generated and optimised; this will make the paper more accessible to a broader audience.

e) Clearly articulate and acknowledge that while TT-TFHE enhances privacy, inference-time privacy does not eliminate all risks.

f) You have already mentioned the challenges of presenting results on ImageNet, but please be more explicit about why this is the case and what could be done to make ImageNet-scale experiments more practical.

**Audience:**

Yes

**Audience Explanation:**

The paper is directly relevant to the privacy-preserving machine learning (PPML) and homomorphic encryption (HE) communities. It addresses a practical and timely challenge, i.e. efficient encrypted inference under TFHE, which is well within TMLR’s scope. With that being said, the audience will be smaller than in other, more established, and popular areas.

Despite being a systems-heavy topic, the paper is structured to be accessible to ML researchers. The authors provide sufficient background on TFHE, TTnets, and the Concrete library, making the paper accessible to readers without deep cryptographic expertise.

**Claims And Evidence:**

Yes

**Claims Explanation:**

The main claims of this paper are tri-fold:

a) TT-TFHE enables practical inference under TFHE on both tabular and small-image datasets;

b) The framework achieves better runtime and memory efficiency than other TFHE-based approaches;

c) The implementation is usable, open, and reproducible within the Concrete environment. The authors provide quantitative comparisons across tabular and image benchmarks (e.g. MNIST, CIFAR-10) and demonstrate consistent performance and memory improvements relative to baseline FHE setups.

---

> ### Author Response · Authors · 2025-11-22
> **Revised Version**
>
> Dear Action Editor,
>
> We submitted the revised version with the following changes:
>
> a) Threat model: Section 4.1
>
> b) Comparative context: Section 2
>
> c) Avoid overstatements such as “fully production-ready” or “first practical FHE inference,”: removed such overstatements
>
> d) Briefly explain how truth-table CNN filters are generated and optimised: Example in Section 3
>
> e) Clearly articulate and acknowledge that while TT-TFHE enhances privacy, inference-time privacy does not eliminate all risks: Limitations in Section 7.1
>
> f) ImageNet: Section 5.2.3
>
> Once these changes are aknowledged we will release the camera ready version with the authors names etc.
>
> We would like to thanks the reviewers for their remarks and comments that helped further improve our work.
>
> The Authors